# Unsupervised Spectral Learning of FSTs

**Raphaël Bailly**     **Xavier Carreras**     **Ariadna Quattoni**
Universitat Politecnica de Catalunya
Barcelona, 08034
rbailly,carreras,aquattoni@lsi.upc.edu

## Abstract

Finite-State Transducers (FST) are a standard tool for modeling paired input-output sequences and are used in numerous applications, ranging from computational biology to natural language processing. Recently Balle et al. [4] presented a spectral algorithm for learning FST from samples of aligned input-output sequences. In this paper we address the more realistic, yet challenging setting where the alignments are unknown to the learning algorithm. We frame FST learning as finding a low rank Hankel matrix satisfying constraints derived from observable statistics. Under this formulation, we provide identifiability results for FST distributions. Then, following previous work on rank minimization, we propose a regularized convex relaxation of this objective which is based on minimizing a nuclear norm penalty subject to linear constraints and can be solved efficiently.

## 1   Introduction

This paper addresses the problem of learning probability distributions over pairs of input-output sequences, also known as transduction problem. A pair of sequences is made of an *input* sequence, built from an input alphabet, and an *output* sequence, built from an output alphabet. Finite State Transducers (FST) are one of the main probabilistic tools used to model such distributions and have been used in numerous applications ranging from computational biology to natural language processing. A variety of algorithms for learning FST have been proposed in the literature, most of them are based on EM optimizations [9, 11] or grammatical inference techniques [8, 6].

In essence, an FST can be regarded as an HMM that generates bi-symbols of combined input-output symbols. The input and output symbols may be generated jointly or independently conditioned on the previous observations. A particular generation pattern constitutes what we call an alignment.

```
GAATTCAG-     GAATTCAG-     GAATTC-AG     GAATTC-AG
| |  || |     | || | |      | | || |      | || | |
GGA-TC-GA     GGAT-C-GA     GGA-TCGA-     GGAT-CGA-
```

To be able to handle different alignments, a special empty symbol $*$ is added to the input and output alphabets. With this enlarged set of bi-symbols, the model is able to generate an input symbol (resp. an output symbol) without an output symbol (resp. input symbol). These special bi-symbols will be represented by the pair $\frac{x}{*}$ (resp. $\frac{*}{y}$). As an example, the first alignment above will correspond to the two possible representations $\frac{GA*ATTCAG*}{G*GA*TC*GA}$ and $\frac{G*AATTCAG*}{GG*A*TC*GA}$. Under this model the probability of observing a pair of un-aligned input-output sequences is obtained by integrating over all possible alignments.

Recently, following a recent trend of work on spectral learning algorithms for finite state machines [14, 2, 17, 18, 7, 16, 10, 5], Balle et al. [4] presented an algorithm for learning FST where the input to the algorithm are samples of *aligned* input-output sequences. As with most spectral methods the core idea of this algorithm is to exploit low-rank decompositions of some *Hankel matrix* representing

the distribution of aligned sequences. To estimate this Hankel matrix it is assumed that the algorithm can sample aligned sequences, i.e. it can directly observe sequences of enlarged bi-symbols.

While the problem of learning FST from fully aligned sequences (what we sometimes refer to as supervised learning) has been solved, the problem of deriving an unsupervised spectral method that can be trained from samples of input-output sequences alone (i.e. where the alignment is hidden) remains open. This setting is significantly more difficult due to the fact that we must deal with two sets of hidden variables: the states and the alignments. In this paper we address this unsupervised setting and present a spectral algorithm that can approximate the distribution of paired sequences generated by an FST without having access to aligned sequences. To the best of our knowledge this is the first spectral algorithm for this problem.

The main challenge in the unsupervised setting is that since the alignment information is not available, the Hankel matrices (as in [4]) can no longer be directly estimated from observable statistics. However, a key observation is that we can nevertheless compute observable statistics that can constraint the coefficients of the Hankel matrix. This is because the probability of observing a pair of un-aligned input-output sequences (i.e. an observable statistic) is computed by summing over all possible alignments; i.e. by summing entries of the hidden Hankel matrix. The main idea of our algorithm is to exploit these constraints and find a Hankel matrix (from which we can directly recover the model) which both agrees on the observed statistics and has a low-rank matrix factorization.

In brief, our main contribution is to show that an FST can be approximated by solving an optimization which is based on finding a low-rank matrix satisfying a set of constraints derived from observable statistics and Hankel structure. We provide sample complexity bounds and some identifiability results for this optimization that show that, theoretically, the rank and the parameters of an FST distribution can be identified. Following previous work on rank minimization, we propose a regularized convex relaxation of the proposed objective which is based on minimizing a nuclear norm penalty subject to linear constraints. The proposed relaxation balances a trade-off between model complexity (measured by the nuclear norm penalty) and fitting the observed statistics. Synthetic experiments show that the performance of our unsupervised algorithm efficiently approximates that of a supervised method trained from fully aligned sequences.

The paper is organized as follows. Section 2 gives preliminaries on FST and spectral learning methods, and establishes that an FST can be induced from a Hankel matrix of observable aligned statistics. Section 3 presents a generalized form of Hankel matrices for FST that allows to express observation constraints efficiently. One can not observe generalized Hankel matrices without assuming access to aligned samples. To solve this problem, Section 4 formulates finding the Hankel matrix of an FST from unaligned samples as rank minimization problem. This section also presents the main theoretical results of the method, as well as a convex relaxation of the rank minimization problem. Section 5 presents results on synthetic data and Section 6 concludes.

## 2 Preliminaries

### 2.1 Finite-State Transducers

**Definition 1.** *A Finite-State Transducer (FST) of rank $d$ is given by:*

- *alphabets $\Sigma_+ = \{x_1, \ldots, x_p\}$ (input), $\Sigma_- = \{y_1, \ldots, y_q\}$ (output)*

- *$\alpha_1 \in \mathbb{R}^d$, $\alpha_\infty \in \mathbb{R}^d$*

- *$\forall x \in \Sigma_+ \cup \{*\}, \forall y \in \Sigma_- \cup \{*\}$, a matrix $M_y^x \in \mathbb{R}^{d \times d}$, with $M_*^* = 0$*

**Definition 2.** *Let $s$ be an input sequence, and let $t$ be an output sequence. An* alignment *of $(s,t)$ is given by a sequence of pairs $\frac{x_1}{y_1} \ldots \frac{x_n}{y_n}$ such that the sequence obtained from $x_1 \ldots x_n$ (resp. $y_1 \ldots y_n$) by removing the empty symbols $*$ equals $s$ (resp. $t$).*

**Definition 3.** *The set of alignments for a pair of sequences $(s,t)$ is denoted $[s,t]$.*

**Definition 4.** *Let $\Sigma = \{\Sigma_+ \cup \{*\}\} \times \{\Sigma_- \cup \{*\}\}$. The set of aligned sequences is $\Sigma^*$. The empty string is denoted $\varepsilon$.*

**Definition 5.** *Let $T$ be an FST, and let $w = \frac{x_1}{y_1} \ldots \frac{x_n}{y_n}$ be an aligned sequence. Then the value of $w$ for the model $T$ is given by:*

$$r_T(w) = \alpha_1^\top M_{y_1}^{x_1} \cdots M_{y_n}^{x_n} \cdot \alpha_\infty$$

**Definition 6.** *Let $(s,t)$ be an i/o (input/output) sequence. Then the value for $(s,t)$ computed by an FST $T$ is given by the sum of the values for all alignments:*

$$r_T((s,t)) = \sum_{\substack{x_1 \ldots x_n \\ y_1 \cdots y_n \in [s,t]}} r_T(\tfrac{x_1}{y_1} \ldots \tfrac{x_n}{y_n})$$

A more complete description of FST can be found in [15].

## 2.2 Computing with an FST

In order to compute the value of a pair of sequences $(s,t)$, one needs to sum over all possible alignments, which is generally exponential in the length of $s$ and $t$. Standard techniques (e.g. the edit distance algorithm) can be applied in order to compute such a value in polynomial time.

**Proposition 1.** *Let $T$ be an FST, $s_{1:n} \in \Sigma_+^*$, $t_{1:m} \in \Sigma_-^*$. The* forward *vector is defined by:*

$$F_{0,j} = \alpha_1^\top M_{t_1}^* \cdots M_{t_j}^*, F_{i,0} = \alpha_1^\top M_*^{s_1} \cdots M_*^{s_i}, F_{i,j} = F_{i-1,j} M_*^{s_i} + F_{i,j-1} M_{t_j}^* + F_{i-1,j-1} M_{t_j}^{s_i}$$

It is then possible to compute $r_T((s,t)) = F_{n,m} \alpha_\infty$ in $O(d^2|s||t|)$. The sum of $r_T$ over all possible values $r_T(\Sigma^*) = \sum_{s \in \Sigma_+, t \in \Sigma_-} r_T((s,t))$ can be computed with the formula

$$r_T(\Sigma^*) = \alpha_1^\top [I_d - M]^{-1} \alpha_\infty$$

where $M = \sum_{x \in \Sigma_+ \cup \{*\}, y \in \Sigma_- \cup \{*\}} M_y^x$.

**Example 1.** *Let us consider a particular subclass of FST: $\Sigma_- = \Sigma_+ = \{0,1\}$, where $M_1^1 = M_0^0 = M_*^0 = M_1^* = 0$.*

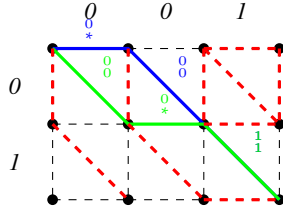

$$\alpha_1 = \begin{pmatrix} 1 \\ 0 \end{pmatrix} \quad M_*^1 = \begin{pmatrix} 1/3 & 0 \\ 0 & 1/6 \end{pmatrix} \quad M_0^* = \begin{pmatrix} 1/6 & 0 \\ 0 & 1/3 \end{pmatrix}$$

$$\alpha_\infty = \begin{pmatrix} 1/4 \\ 0 \end{pmatrix} \quad M_0^0 = \begin{pmatrix} 0 & 1/4 \\ 0 & 0 \end{pmatrix} \quad M_1^1 = \begin{pmatrix} 0 & 0 \\ 1/2 & 0 \end{pmatrix}$$

*The FST $A$ satisfies the constraints.*

*Let us draw all the paths for the i/o sequence $(01,001)$. The dashed red edges are discarded because of the constraints. Thus, there are only two different non-zero paths, corresponding to $\frac{0*1}{001}$ (in green) and $\frac{*01}{001}$ (in blue).*

*Let us consider the model $A$ which satisfies the constraints above. One has $r_A(\frac{0*1}{001}) = 1/96, r_A(\frac{*01}{001}) = 1/192$ and, as those two aligned sequences are the only possible alignments for $(01,001)$, one has $r_A((01,001)) = 1/64$. It is possible to check that $r_A(\Sigma^*) = 1$, thus the model computes a probability distribution.*

## 2.3 Hankel Matrices

Let us recall some basic definitions and properties.

Let $\Sigma$ be an alphabet, $U \subset \Sigma^*$ a set of prefixes, $V \subset \Sigma^*$ a set of suffixes. $U$ is said to be *prefix-closed* if $uv \in U \Rightarrow u \in U$. $V$ is said to be *suffix-closed* if $uv \in V \Rightarrow v \in V$.

Let us denote $U\Sigma$ the set $U \cup \{u\frac{x}{y} | u \in U, \frac{x}{y} \in \Sigma\}$. Let us denote $\Sigma V$ the set $V \cup \{\frac{x}{y}v | v \in V, \frac{x}{y} \in \Sigma\}$.

A Hankel matrix on $U$ and $V$ is a matrix with rows corresponding to elements $u \in U$ and columns corresponding to elements $v \in V$, which satisfies $uv = u'v' \rightarrow H(u,v) = H(u',v')$.

**Definition 7.** *Let $H$ a Hankel matrix for $U\Sigma$ and $\Sigma V$. One supposes that $\varepsilon \in U$ and $\varepsilon \in V$. One then defines the partial Hankel matrices defined for $u \in U$ and $\in V$:*

$$H_\varepsilon(u,v) = H(u,v) \quad , \quad H_y^x(u,v) = H(u,\tfrac{x}{y}v) \quad , \quad H_1(v) = H(\varepsilon,v) \quad , \quad H_\infty(u) = H(u,\varepsilon)$$

The main result that we will be using is the following:

**Proposition 2.** *Let $H$ a Hankel matrix for $U\Sigma$ and $\Sigma V$. One supposes that $U$ is prefix-closed, $V$ is suffix-closed, and that $rank(H_\varepsilon) = rank(H)$. Then the WA defined by*

$$\alpha_1^\top = H_1^\top H_\varepsilon^+ \quad , \quad \alpha_\infty = H_\infty \quad , \quad M_y^x = H_y^x H_\varepsilon^+$$

*computes a mapping $f$ such that $\forall u \in U, \forall v \in V, f(uv) = H_\varepsilon(u,v)$.*

We will not give a proof of this result, as a more general result is seen further. The rank equality comes from the fact that the WA defined above has the same rank as $H_\varepsilon$, and that the rank of a mapping $f$ which satisfies $f(uv) = H(u,v)$ is at least the rank of $H$. The following example shows that the prefix and suffix closeness are necessary conditions.

**Example 2.** *Let us consider the following Hankel over the set of prefixes $U\Sigma$ and the set of suffixes $\Sigma V$ with $U = \{\varepsilon, \frac{a^3}{b^3}\}$, and $V = \{\varepsilon, \frac{a^3}{b^3}\}$.*

$$
H_\varepsilon = \begin{matrix} \varepsilon \\ \frac{a^3}{b^3} \end{matrix} \begin{pmatrix} \varepsilon & \frac{a^3}{b^3} \\ 0 & 0 \\ 0 & 1/4 \end{pmatrix}, \quad
H = \begin{matrix} \varepsilon \\ \frac{a}{b} \\ \frac{a^3}{b^3} \\ \frac{a^4}{b^4} \end{matrix} \begin{pmatrix} \varepsilon & \frac{a}{b} & \frac{a^3}{b^3} & \frac{a^4}{b^4} \\ 0 & 0 & 0 & 0 \\ 0 & 0 & 0 & 0 \\ 0 & 0 & 1/4 & 1/4 \\ 0 & 0 & 1/4 & 1/4 \end{pmatrix}, \quad
H' = \begin{matrix} \frac{a^2}{b^2} \\ \frac{a^3}{b^3} \\ \frac{a^4}{b^4} \end{matrix} \begin{pmatrix} \frac{a^2}{b^2} & \frac{a^3}{b^3} & \frac{a^4}{b^4} \\ 0 & 0 & 1/4 \\ 0 & 1/4 & 1/4 \\ 1/4 & 1/4 & 1/4 \end{pmatrix}
$$

*One has $\varepsilon \in U$ and $\varepsilon \in V$, and also $rank(H_\varepsilon) = rank(H) = 1$, thus the computed WA is rank 1. Such a WA cannot compute a mapping such that $r_T(\epsilon) = 0$ and $r_T(\frac{a^6}{b^6}) = 1/4$. The complete Hankel matrix has at least rank 7.*

## 3 Inducing FST from Generalized Hankel Matrices

Proposition 2 tells us that if we had access to certain sub-blocks of the Hankel matrix for aligned sequences we could recover the FST model. However, we do not have access to the hidden alignment information: we only have access to the statistics $p(s,t)$, which we will call *observations*. One natural idea would be to search for a Hankel matrix that agrees with the observations. To do so, we introduce *observable constraints*, which are linear constraints of the form $p(s,t) = \sum_{\substack{x_1...x_n \\ y_1...y_n} \in [s,t]} r_T(\begin{smallmatrix} x_1 & ... & x_n \\ y_1 & ... & y_n \end{smallmatrix})$, where $r_T(\begin{smallmatrix} x_1 & ... & x_n \\ y_1 & ... & y_n \end{smallmatrix})$ is computed from the Hankel.

From a matrix satisfying the hypothesis of Proposition 2 *and* the observation constraints, one can derive an FST computing a mapping which agrees on the observations.

Given an i/o sequence $(s,t)$, the size of $[s,t]$ (hence the size of the Hankel matrix) is in general exponential in the size of $s$ and $t$. In order to overcome that problem when considering the observation constraints, one will consider aggregations of rows and columns, corresponding to sets of prefixes and suffixes. Obviously, the definition of a Hankel matrix must be extended to this case.

One will denote by $\mathscr{P}(A)$ the set of subsets of $A$.

**Definition 8.** *Let $u, u' \in \mathscr{P}(\Sigma^*)$. The set $uu'$ is defined by $uu' = \{ww' | w \in u, w' \in u'\}$.*

We will denote sets of alignments as follows: $\frac{x_{1:n}}{y_{1:n}}$ will denote the set $\{\frac{x_{1:n}}{y_{1:n}}\}$, which contains a single aligned sequence; then $\frac{x_{1:n}}{y_{1:n}}[s,t]$ will denote the set $\{\frac{x_{1:n}x_{n+1:n+k}}{y_{1:n}y_{n+1:n+k}} \mid \frac{x_{n+1:n+k}}{y_{n+1:n+k}} \in [s,t]\}$, which extends $\{\frac{x_{1:n}}{y_{1:n}}\}$ with all ways of aligning $(s,t)$. We will also use $[s,t]^{x_{1:n}}_{y_{1:n}}$ analogously.

**Definition 9.** *A generalized prefix (resp. generalized suffix) is the empty set or a set of the form $[s,t]^{x_{1:n}}_{y_{1:n}}$ (resp. $\frac{x_{1:n}}{y_{1:n}}[s,t]$), where the aligned part is possibly empty.*

### 3.1 Generalized Hankel

One needs to extend the definition of a Hankel matrix to the generalized prefixes and suffixes.

**Definition 10.** *Let $U$ be a set of generalized prefixes, $V$ be a set of generalized suffixes. Let $H$ be a matrix indexed by $U$ (in rows) and $V$ (in columns). $H$ is a generalized Hankel matrix if it satisfies:*

$$\forall \bar{u}, \bar{u}' \subset U, \forall \bar{v}, \bar{v}' \subset V, \quad \biguplus_{u \in \bar{u}, v \in \bar{v}} uv = \biguplus_{u' \in \bar{u}', v' \in \bar{v}'} u'v' \Rightarrow \sum_{u \in \bar{u}, v \in \bar{v}} H(u,v) = \sum_{u' \in \bar{u}', v' \in \bar{v}'} H(u'v')$$

*where $\biguplus$ is the disjoint union.*

In particular, if $U$ and $V$ are sets of regular prefixes and suffixes, this definition encompasses the regular definition for a Hankel matrix.

**Definition 11.** *Let $U$ be a set of generalized prefixes. $U$ is* prefix-closed *if*

$$[s,t]_{y_{1:n}}^{x_{1:n}} \in U \Rightarrow [s,t]_{y_{1:n-1}}^{x_{1:n-1}} \in U$$

$$[s_{1:n}, t_{1:k}] \in U \Rightarrow [s_{1:n-1}, t_{1:k}]_*^{s_n}, [s_{1:n}, t_{1:k-1}]_{t_k}^*, [s_{1:n-1}, t_{1:k-1}]_{t_k}^{s_n} \in U$$

**Definition 12.** *Let $V$ be a set of generalized suffixes. $V$ is* suffix-closed *if*

$$_{y_{1:n}}^{x_{1:n}}[s,t] \in V \Rightarrow {}_{y_{2:n}}^{x_{2:n}}[s,t] \in V$$

$$[s_{1:n}, t_{1:k}] \in V \Rightarrow {}_*^{s_1}[s_{2:n}, t_{1:k}], {}_{t_1}^*[s_{1:n}, t_{2:k}], {}_{t_1}^{s_1}[s_{2:n}, t_{2:k}] \in V$$

**Definition 13.** *Let $U$ be a set of generalized prefixes, $V$ be a set of generalized suffixes. The right-operator completion of $U$ is the set $U\Sigma = U \cup \{u_y^x | u \in U, \frac{x}{y} \in \Sigma\}$. The left-operator completion of $V$ is the set $\Sigma V = V \cup \{_y^x v | v \in V, \frac{x}{y} \in \Sigma\}$.*

A key result is the following, which is analogous to Proposition 2 for generalized Hankel matrices:

**Proposition 3.** *Let $U$ and $V$ be two sets of generalized prefixes and generalized suffixes. Let $H$ be a Hankel matrix built from $U\Sigma$ and $\Sigma V$. One supposes that $rank(H_\varepsilon) = rank(H)$, $U$ is* prefix-closed *and $V$ is* suffix-closed. *Then the model $A$ defined by $\alpha_1^\top = H_1^\top H_\varepsilon^+, \alpha_\infty = H_\infty, M_y^x = H_y^x H_\varepsilon^+$ computes a mapping $r_A$ such that*

$$\forall u \in U, \forall v \in V, r_A(uv) = H_\varepsilon(u,v)$$

*Proof.* The proof can be found in the Appendix. ☐

Let $S$ be a sample of unaligned sequences. Let $\text{pref}_{S,in}$ (resp. $\text{pref}_{S,out}$, $\text{suff}_{S,in}$, $\text{suff}_{S,out}$) be the prefix (resp. suffix) closure of input (resp. output) strings in $S$. Let $U = \{[s,t]\}_{s \in \text{pref}_{S,in}, t \in \text{pref}_{S,out}}$ and $V = \{[s,t]\}_{s \in \text{suff}_{S,in}, t \in \text{suff}_{S,out}}$. The sets $U$ and $V$ contain all the observed pairs of unaligned sequences, and one can check that the sizes of $U\Sigma$ and $\Sigma V$ are polynomial in the size of $S$.

**Example 3.** *Let us continue with the same model $A$ as in Example 1. Let us now consider the prefixes $\epsilon, \frac{0}{0}$ and the suffixes $\epsilon, \frac{1}{1}$. The Hankel matrices will be:*

$$H_1 = \begin{pmatrix} 1/4 \\ 0 \end{pmatrix} H_\infty = \begin{pmatrix} 1/4 \\ 0 \end{pmatrix} H_\varepsilon = \begin{pmatrix} 1/4 & 0 \\ 0 & 1/32 \end{pmatrix} H_{\frac{*}{0}} = \begin{pmatrix} 1/24 & 0 \\ 0 & 1/96 \end{pmatrix}$$

$$H_{\frac{1}{*}} = \begin{pmatrix} 1/12 & 0 \\ 0 & 1/192 \end{pmatrix} H_{\frac{0}{0}} = \begin{pmatrix} 0 & 1/32 \\ 0 & 0 \end{pmatrix} H_{\frac{1}{1}} = \begin{pmatrix} 0 & 0 \\ 1/32 & 0 \end{pmatrix}$$

*and one finally gets the model $A'$ defined by:*

$$\alpha_1 = \begin{pmatrix} 1 \\ 0 \end{pmatrix} \alpha_\infty = \begin{pmatrix} 1/4 \\ 0 \end{pmatrix} M_{\frac{1}{*}} = \begin{pmatrix} 1/3 & 0 \\ 0 & 1/6 \end{pmatrix}$$

$$M_{\frac{*}{0}} = \begin{pmatrix} 1/6 & 0 \\ 0 & 1/3 \end{pmatrix} M_{\frac{0}{0}} = \begin{pmatrix} 0 & 1 \\ 0 & 0 \end{pmatrix} M_{\frac{1}{1}} = \begin{pmatrix} 0 & 0 \\ 1/8 & 0 \end{pmatrix}$$

*One can easily check that $A'$ computes the same probability distribution than $A$.*

## 4  FST Learning as Non-convex Optimization

Proposition 3 shows that FST models can be recovered from generalized Hankel matrices. In this section we show how FST learning can be framed as an optimization problem where one searches for a low-rank generalized Hankel matrix that agrees with observation constraints derived from a sample. We assume here that $p$ is a probability distribution over i/o sequences.

We will denote by $\boldsymbol{z} = (p([s,t]))_{s \in \Sigma_+^*, t \in \Sigma_-^*}$ the vector built from observable probabilities, and by $\boldsymbol{z}_S = (p_S([s,t]))_{s \in \Sigma_+^*, t \in \Sigma_-^*}$ the set of empirical observable probabilities, where $p_S$ is the frequency deduced from an i.i.d. sample $S$.

Let $\vec{H}$ be the vector describing the coefficients of $H$. Let $K$ be the matrix such that $K\vec{H} = 0$ represents the Hankel constraints (cf. definition 10). Let $O$ be the matrix such that $O\vec{H} = z$ represents the observation constraints (i.e. $\sum H([s,t],\epsilon) = p([s,t])$).

The optimization task is the following:

$$
\begin{aligned}
\underset{H}{\text{minimize}} \quad & rank(H) \\
\text{subject to} \quad & \|O\vec{H} - z\|_2 \leq \mu \\
& K\vec{H} = \mathbf{0} \\
& \|H\|_2 \leq 1.
\end{aligned}
\tag{1}
$$

The condition $\|H\|_2 \leq 1$ is necessary to bound the set of possible solutions. In particular, if $H$ is the Hankel matrix of a probability distribution, the condition is satisfied: one has $\|H\|_1 \leq 1$ as each column is a probability distribution, as well as $\|H\|_F \leq 1$ and thus $\|H\|_2 \leq \sqrt{\|H\|_1 \|H\|_F} \leq 1$. Let us remark that the set of matrices satisfying the constraints is a compact set.

Let us denote $\mathcal{H}_\mu$ the class of Hankel matrices solutions of (1) for a given $\mu$, where $z$ represents the $p(u_i v_j)$. Let us denote $\mathcal{H}_\mu^S$ the class of Hankel matrices solutions of (1) for a given $\mu$, where $z_S$ represents $p_S(u_i v_j)$, the observed frequencies in a sample $S$ i.i.d. with respect to $p$.

**Proposition 4.** *Let $p$ be a distribution over i/o sequences computed by an FST. There exists $U$ and $V$ such that any solution of $\mathcal{H}_0$ leads to an FST*

$$
\alpha_1^\top = H_1^\top H_\varepsilon^+, \alpha_\infty = H_\infty, M_y^x = H_y^x H_\varepsilon^+
$$

*which computes $p$.*

*Proof.* The proof can be found in the Appendix. $\qquad\square$

## 4.1 Theoretical Properties

We now present the main theoretical results related to the optimization problem (1). The first one concerns the rank identification, while the second one concerns the consistency of the method.

**Proposition 5.** *Let $p$ be a rank d distribution computed by an FST. There exists $\mu_2$ such that for any $\delta > 0$ and any i.i.d. sample $S$*

$$
|S| > \left( \frac{2 + \sqrt{8 \log(1/\delta)}}{\mu_2} \right)^2
$$

*implies that any $H \in \mathcal{H}_\mu^S$ solution of (1) with $\mu = \frac{1+\sqrt{2\log(1/\delta)}}{\sqrt{|S|}}$ leads to a rank-d FST with probability at least $1 - \delta$.*

*Proof.* The proof can be found in the Appendix. $\qquad\square$

**Proposition 6.** *Let $p$ be a rank d distribution computed by an FST. There exists $\mu_\epsilon$ such that for any $\delta > 0$ and any i.i.d. sample $S$*

$$
|S| > \left( \frac{1 + \sqrt{2 \log(1/\delta)}}{\mu_\epsilon} \right)^2
$$

*implies that for any $H \in \mathcal{H}_\mu^S$ solution of (1) with $\mu = \frac{1+\sqrt{2\log(1/\delta)}}{\sqrt{|S|}}$ leading to a model $A_s$ there exists a model $A$ computing $p$ such that*

$$
|A, A_S|_\infty \leq O(\frac{d^2 \epsilon}{\sigma_p^3})
$$

*where $|A, A_S|_\infty$ is the maximum distance for model parameters, and $\sigma_p$ is a non-zero parameter depending on $p$.*

*Proof.* The proof can be found in the Appendix. $\qquad\square$

**Example 4.** *This continues Examples 1 and 3. Let us first remark that, among all the values used to build the Hankel matrices in Example 3, some of them correspond to observable statistics, as there is only one possible alignment for them. Conversely, the exact values of $r_M \binom{0*1}{001}$ and $r_M \binom{011}{0*1}$ are not observable. Then the rank minimization objective is not sufficient, as it allows any value for those variables.*

*Let us consider now larger sets of prefixes and suffixes: $\varepsilon, \frac{*}{0}, \frac{0}{0}$ and $\varepsilon, \frac{1}{*}, \frac{1}{1}$. One then has*

$$H_\epsilon = \begin{pmatrix} 1/4 & 1/12 & 0 \\ 1/24 & ? & 0 \\ 0 & 0 & 1/32 \end{pmatrix} H_{\frac{1}{*}} = \begin{pmatrix} 1/12 & 1/36 & 0 \\ ? & ? & 0 \\ 0 & 0 & ? \end{pmatrix} H_{\frac{1}{1}} = \begin{pmatrix} 0 & 0 & 0 \\ 0 & 0 & 0 \\ 1/32 & ? & 0 \end{pmatrix}$$

*We want to minimize the rank of $H'$ subject to the constraints $O$ and $K$:*

$$H' = \begin{pmatrix} H_\varepsilon \\ H_{\frac{1}{*}} \\ H_{\frac{1}{1}} \end{pmatrix} = (h_{ij}), \quad O = \begin{cases} h_{11} = 1/4 & \vdots \\ h_{12} = 1/12 & \\ h_{13} = 0 & h_{63} + h_{92} = 1/64 \\ h_{21} = 1/24 & \vdots \end{cases}, \quad K = \begin{cases} h_{12} = h_{41} & h_{23} = h_{81} \\ h_{13} = h_{71} & h_{32} = h_{61} \\ h_{22} = h_{51} & h_{33} = h_{91} \end{cases}$$

*The relation $h_{63} + h_{92} = 1/64$ is due to fact that $r_A \binom{0*1}{001} + r_A \binom{*01}{001} = r_A ((01, 001)) = 1/64$. One has $h_{22} = h_{51}$ as they both represent $p\binom{*1}{0*}$. It is clear that $H'$ has a rank greater or equal than 2. The only way to reach rank 2 under the constraints is*

$$h_{22} = h_{51} = 1/72, h_{52} = 1/216, h_{63} = 1/192, h_{92} = 1/96$$

*Thus, the process of rank minimization under linear constraints leads to one single model, which is identical to the original one. Of course, in the general case, the rank minimization objective may lead to several models.*

## 4.2  Convex Relaxation

The problem as it is stated in (1) is NP-hard to solve in the size of the Hankel matrix, hence impossible to deal with in practical cases. One can solve instead a convex relaxation of the problem (1), obtained by replacing the rank objective by the nuclear norm. The relaxed optimization statement is then the following:

$$\begin{aligned} \underset{H}{\text{minimize}} \quad & \|H\|_* \\ \text{subject to} \quad & \|O\vec{H} - \boldsymbol{z}\|_2 \le \mu \\ & K\vec{H} = \boldsymbol{0} \\ & \|H\|_2 \le 1. \end{aligned} \tag{2}$$

This type of relaxation has been used extensively in multiple settings [13]. The nuclear norm $\|\|_*$ plays the same role than the $\|\|_1$ norm in convex relaxations of the $\|\|_0$ norm, used to reach sparsity under linear constraints.

## 5  Experiments

We ran synthetic experiments using samples generated from random FST with input-output alphabets of size two. The main goal of our experiments was to compare our algorithm to a *supervised* spectral algorithm for FST that has access to alignments. In both methods, the Hankel was defined for prefixes and suffixes up to length 1. Each run consists of generating a target FST, and generating $N$ aligned samples according to the target distribution. These samples were directly used by the supervised algorithm. Then, we removed the alignment information from each sample (thus obtaining a pair of unaligned strings), and we used them to train an FST with our general learning algorithm, trying different values for a $C$ parameter that trades-off the nuclear norm term and the observation term. We ran this experiment for 8 target models of 5 states, for different sampling sizes. We measure the $L_1$ error of the learned models with respect to the target distribution on all unaligned pairs of strings whose sizes sum up to 6. We report results for geometric averages.

In addition, we ran two additional methods. First a *factorized* method that assumes that the two sequences are generated independently, and learns two separate weighted automata using a spectral

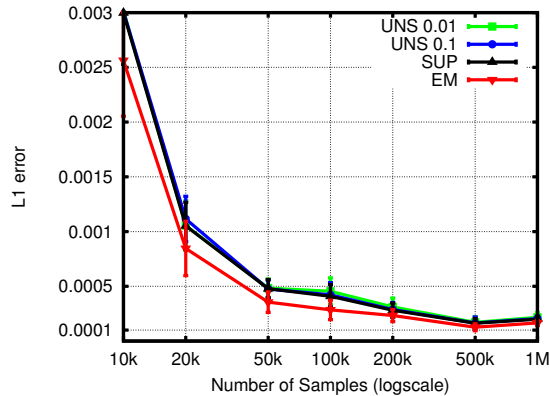

Figure 1: Learning curves for different methods: SUP, supervised; UNS, unsupervised with different regularizers; EM, expectation maximization. The curves are averages of L1 error for random target models of 5 states.

method. Its performance is very bad, with L1 error rates at ~0.08, which confirms that our target models generate highly dependent sequence pairs. This baseline result also implies that the rest of the methods can learn the dependencies between paired strings. Second, we ran an *Expectation Maximization* algorithm (EM).

Figure 1 shows the performance of the learned models with respect to the number of samples. Clearly, our algorithm is able to estimate the target distribution and gets close to the performance of the supervised method, while making use of much simpler statistics. EM performed slightly better than the spectral methods, but nonetheless at the same levels of performance.

One can find other experimental results for the unsupervised spectral method in [1], where it is shown that, under certain circumstances, unsupervised spectral method can perform better than supervised EM. Though the framework (unsupervised learning of PCFGs) is not the same, the method is similar and the optimization statement is identical.

# 6 Conclusion

In this paper we presented a spectral algorithm for learning FST from unaligned sequences. This is the first paper to derive a spectral algorithm for the unsupervised FST learning setting. We prove that there is theoretical identifiability of the rank and the parameters of an FST distribution, using a rank minimization formulation. However, this problem is NP-hard, and it remains open whether there exists a polynomial method with identifiability results. Classically, rank minimization problems are solved with convex relaxations such as the nuclear norm minimization we have proposed. In experiments, we show that our method is comparable to a fully supervised spectral method, and close to the performance of EM.

Our approach follows a similar idea to that of [3] since both works combine classic ideas from spectral learning with classic ideas from low rank matrix completion. The basic idea is to frame learning of distributions over structured objects as a low-rank matrix factorization subject to linear constraints derived from observable statistics. This method applies to other grammatical inference domains, such as unsupervised spectral learning of PCFGs ([1]).

**Acknowledgments**

We are grateful to Borja Balle and the anonymous reviewers for providing us with helpful comments. This work was supported by a Google Research Award, and by projects XLike (FP7-288342), ERA-Net CHISTERA VISEN, TACARDI (TIN2012-38523-C02-02), BASMATI (TIN2011-27479-C04-03), and SGR-LARCA (2009-SGR-1428). Xavier Carreras was supported by the Ramón y Cajal program of the Spanish Government (RYC-2008-02223).

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
