[Supplementary Material · supp_466.pdf]

# 7 Appendix

## 7.1 Proof of the Proposition 3

In order to prove that result, one needs some intermediate results. Let $H_U$ (resp. $H_{U_y^x}$) be the submatrix of $H$ corresponding to prefixes in $U$ (resp. of the form $u_y^x$ with $u \in U$). Let $H_V$ (resp. $H_{_y^x V}$) be the submatrix of $H$ corresponding to suffixes in $V$ (resp. of the form $_y^x v$ with $v \in V$).

**Lemma 1.** *Let $\boldsymbol{u}$ and $\boldsymbol{v}$ be two vectors such that $\boldsymbol{u}^\top H_\varepsilon = \boldsymbol{v}^\top H_\varepsilon$. Then, for $x \in \Sigma_+ \cup \{*\}, y \in \Sigma_- \cup \{*\}$, one has $\boldsymbol{u}^\top H_{_y^x} = \boldsymbol{v}^\top H_{_y^x}$.*

*Proof.* $H_\varepsilon$ is a submatrix of $H_U$ with the same rank.

Let $\boldsymbol{u}$ and $\boldsymbol{v}$ be two vectors such that $\boldsymbol{u}^\top H_\varepsilon = \boldsymbol{v}^\top H_\varepsilon$, then $\boldsymbol{u}^\top H_U = \boldsymbol{v}^\top H_U$ because $H_\varepsilon$ and $H_U$ have the same rank. Thus, as each $H_{_y^x}$ is a submatrix of $H_U$, one has $\boldsymbol{u}^\top H_{_y^x} = \boldsymbol{v}^\top H_{_y^x}$. $\qquad\square$

**Lemma 2.** *Let $u \in U\Sigma$. Then the vector*

$$\sum_{\substack{x_1 \cdots x_n \\ y_1 \cdots y_n \in u}} (H_{_{y_n}^{x_n}})^\top \cdots (H_{_{y_1}^{x_1}} H_\varepsilon^+)^\top ((H_\varepsilon^+)^\top H_1)$$

*is the row of $H_V$ corresponding to the prefix $u$. In particular, if $u \in U$, the vector is equal to the row of $H_\varepsilon$ corresponding to the prefix $u$.*

*Proof.* By induction. $H_1$ is the row of $H_V$ corresponding to $\varepsilon$.

1) Let us suppose that $u = u'_y^x$. Because $U\Sigma$ is prefix-closed, one has $u' \in U\Sigma$. Let $z'$ be the row of $H_V$ corresponding to $u'$. $(H_\varepsilon^+)^\top z'$ represents a decomposition of $z'$ in terms of rows of $H_\varepsilon$. The vector $(H_{_y^x})^\top (H_\varepsilon^+)^\top z$ is the same linear combination of rows of $H_{_y^x V}$, and by rank equality is the same as the row of $H_{_y^x V}$ corresponding to $u'$. Because $H$ is a Hankel matrix, it is equal to the row of $H_V$ corresponding to $u'_y^x = u$.

2) Let us suppose that $u = [s_{1:n}, t_{1:k}]$. Then $u_1 = [s_{1:n-1}, t_{1:k}]_*^{s_n} \in U\Sigma$, $u_2 = [s_{1:n}, t_{1:k-1}]_{t_k}^* \in U\Sigma$, $u_3 = [s_{1:n-1}, t_{1:k-1}]_{t_k}^{s_n} \in U\Sigma$. With the same argument as before applied to $u_1$, $u_2$ and $u_3$, and because $H$ is Hankel, one has the result. $\qquad\square$

One has then the symmetric result for the suffixes.

**Lemma 3.** *Let $v \in \Sigma V$. Then the vector*

$$\sum_{\substack{x_1 \cdots x_n \\ y_1 \cdots y_n \in v}} (H_{_{y_1}^{x_1}}) \cdots (H_{_{y_n}^{x_n}} H_\varepsilon^+)(H_\infty)$$

*is the column of $H_U$ corresponding to the suffix $v$. In particular, if $v \in V$, the vector is equal to the column of $H_\varepsilon$ corresponding to the suffix $v$.*

*Proof.* It is just the symmetric case of the previous lemma. $\qquad\square$

### 7.1.1 Proof of the Proposition 3

Let $u \in U, v \in V$. Let $H_u$ be the row of $H_\varepsilon$ corresponding to $u$, $H_v$ the column of $H_\varepsilon$ corresponding to $v$. One then has, by Lemma 2 and Lemma 3, $r_M(uv) = H_u^\top H_\varepsilon^+ H_v$ The vector $H_\varepsilon^+ H_v$ represents a decomposition of $H_v$ equivalent to the vector $\mathbf{1}_v$. Then $r_M(uv) = H_u^\top \mathbf{1}_v = H_\varepsilon(u, v)$. $\qquad\square$

## 7.2 Proof of the Proposition 4

**Definition 14.** *Let $p$ be a distribution over i/o sequences computed by an FST. Let $rank(p)$ be the minimal integer $d$ such that there exist an FST with $d$ states computing $p$. Let $\mathcal{V}_p$ be the class of parameters for all rank-$d$ FSTs over bi-sequences which compute the same distribution over i/o sequences as $p$.*

**Definition 15.** *An affine variety is the set of solutions of a (maybe infinite) polynomial equation system:*

$$\begin{cases} P_1(X_1,\ldots,X_n) = 0 \\ \quad\vdots \end{cases}$$

**Lemma 4.** *Let $p$ be a rank $d$ distribution over bi-sequences computed by an FST. Then $\mathcal{V}_p$ is an affine variety.*

*Proof.* Let $A$ be a $d$-state FST. The value computed by $A$ for a given i/o sequence $(s,t)$ is a polynom in its parameter denoted $P_{(s,t)}$. Thus, the set of parameters corresponding to $d$-state FST computing a given value $p((s,t))$ for $(s,t)$ is an affine variety defined by $\{(X_1,\ldots,X_n)|P_{(s,t)}-p((s,t)) = 0\}$, and $\mathcal{V}_p$ is the affine variety defined by: $\bigcap_{(s_i,t_j)\in\Sigma_+\times\Sigma_-}\{(X_1,\ldots,X_n)|P_{(s_i,t_j)}-p((s_i,t_j)) = 0\}$. $\square$

**Lemma 5.** *Let $p$ be a rank $d$ distribution over bi-sequences computed by an FST. Then there exists a finite set $G_p$ of i/o sequences, such that $\mathcal{V}_p = \bigcap_{(s_i,t_j)\in G_p}\{(X_1,\ldots,X_n)|P_{(s_i,t_j)}-p((s_i,t_j)) = 0\}$. Such a set $G_p$ is called a* generative set *for $p$.*

*Proof.* The ring $\mathbb{R}[X_1,\ldots,X_n]$ is Noetherian, in particular the sequence $I_k = \bigcap_{k'\leq k}\{(X_1,\ldots,X_n)|P_{(s_{i_{k'}},t_{j_{k'}})}(X_1,\ldots,X_n) - p((s_{i_{k'}},t_{j_{k'}})) = 0\}$ is stationary. One has $\mathcal{V}_p = \bigcup_n I_n = \bigcup_{n\leq N} I_n$ for a certain $N$. One can take $G_p = \bigcup_{n\leq N}(s_{i_n},t_{j_n})$. $\square$

**Corollary 1.** *Let $p$ be a rank $d$ distribution over i/o sequences computed by an FST. Let $G_p$ be a generative set for $p$. Let $A$ be an FST of rank $\leq d$. One then has:*

$$r_A|_{G_p} = p|_{G_p} \Leftrightarrow r_A = p$$

### 7.2.1  Proof of Proposition 4

*Proof.* Let $p$ be a rank $d$ distribution over i/o sequences computed by an FST. Let $G_p$ be a generative set for $p$. Let $U_0$ (resp. $V_0$) be the prefix-closure (resp. suffix-closure) of $G_p$. Let $U_{i+1} = U_i\Sigma$, $U = U_{d+1}$ and $V_{i+1} = \Sigma V_i$, $V = V_{d+1}$. Let $H_i$ be the minimum rank Hankel matrix over $U_i$ and $V_i$, and let $H$ be a minimum rank Hankel matrix over $U$ and $V$. With Corollary 1 and Proposition 3, it is sufficient to prove that $rank(H_d) = rank(H) = d$. As the Hankel matrix of $p$ fulfills the hypothesis, one has $rank(H) \leq d$. Among the family of $(d + 1)$ couples $(H_0, H_1),\ldots(H_d, H)$, one of them satisfies $rank(H_i) = rank(H_{i+1})$, because otherwise $rank(H_i)$ would take $d + 2$ different values between $0$ and $d$. Thus, the FST computed from $H_{i+1}$ agrees on $G_p$ with $p$ by Proposition 3, and by Corollary 1, as $G_p \subset U \times V$, this FST computes $p$. By minimality of the rank, one has $rank(H_i) = rank(H_{i+1}) = d$, and thus $rank(H_d) = rank(H) = d$. $\square$

### 7.3  Proof of the Proposition 5

**Lemma 6.** *Let $p$ be a rank $d$ distribution computed by an FST. Let $U$ and $V$ be such as in Proposition 4. There exists $\sigma > 0$ such $H \in \mathcal{H}_0 \Rightarrow \sigma_d(H_\varepsilon) \geq \sigma$, where $\sigma_d(H_\varepsilon)$ is the $d$-th singular vaue of $H_\varepsilon$.*

*Proof.* For $\mu = 0$, the rank minimization is equivalent to $rank(H) \leq d$, thus the set $\mathcal{H}_0$ of the solutions of (1) is a closed bounded set, thus compact. Suppose that the assumption is false, this means, by compacity, that one can find a sequence $H_n$ such that $\sigma_d(H_{n\varepsilon})$ converges towards a matrix $H_\omega$ such that $\sigma_d(H_{\omega\varepsilon}) = 0$ by continuity of singular values. As $H_\omega \in \mathcal{H}_0$, The FST obtained from $H_\omega$ computes $p$, which contradicts the fact that $rank(H_{\omega\varepsilon}) = d$ (cf. proof of Proposition 4). $\square$

**Lemma 7.** *Let $p$ be a distribution computed by a rank $d$ FST. Let $U$ and $V$ be such as in Proposition 4. Let $\sigma$ be as in Lemma 6. There exists $\mu_2$ such that $H \in \mathcal{H}_{\mu_2} \Rightarrow \sigma_d(H_\varepsilon) > \sigma/2$.*

*Proof.* Suppose the assumption is false: there exists a convergent sequence of Hankel matrices $H_n \in \mathcal{H}_{1/n}$ such that $\sigma_d(H_{n\varepsilon}) < \sigma/2$, and whose limit is $M_\omega$. One then has $H_\omega \in \mathcal{H}_0$, and $\sigma_d(H_{\omega\varepsilon}) \leq \sigma/2$ by continuity, which contradicts Lemma 6. $\square$

In particular, this implies that, for a certain $\mu_2$, all the solutions $\mathcal{H}_{\mu_2}$ of (1) will be such that $H_\varepsilon$ is rank $d$, thus $\mathcal{H}_{\mu_2}$ is compact.

**Lemma 8.** *Let $p$ be a rank $d$ distribution computed by an FST. Let $U$ and $V$ be such as in Proposition 6. For all $\epsilon > 0$ there exists $\mu_\epsilon$ such that $H \in \mathcal{H}_{\mu_\epsilon} \Rightarrow min_{H_0 \in \mathcal{H}_0}(\|H - H_0\|_F) \le \epsilon$.*

*Proof.* Let us consider $\mu_\epsilon < \mu_2$, $\mu_2$ beeing as in Lemma 7. The rank minimization is equivalent to $rank(H) \le d$, thus the set $\mathcal{H}_{\mu_\epsilon}$ is compact. Let us suppose that the assumption is false, and that there exists a sequence $\mathcal{H}_n$ such that $H_n \in \mathcal{H}_{1/n}$ and $min_{H_0 \in \mathcal{H}_0}(\|H_n - H_0\|_F) > \epsilon$. The limit $H_\omega$ belongs to $\mathcal{H}_0$ and satisfies $min_{H_0 \in \mathcal{H}_0}(\|H_\omega - H_0\|_F) \ge \epsilon$ which is contradictory. $\qquad\square$

**Lemma 9.** *Let $p$ be a rank $d$ distribution computed by an FST. Let $U$ and $V$ be such as in Proposition 4. Let $\delta > 0$ be a confidence parameter. Let $S$ be an i.i.d. sample of size $N$, drawn with respect to $p$. Let $z_S = (p_S([s,t]))_{[s,t] \in U}$ be the vector of frequencies in the sample $S$, and let $z = (p([s,t]))_{[s,t] \in U}$. One has, with probability a least $1 - \delta$:*

$$\|z - z_S\|_2 < \frac{1 + \sqrt{2 \log(1/\delta)}}{\sqrt{N}}$$

*Proof.* Let $S_i$ be a sample differing from $S$ for the $i$-th entry. One has $\|z_S - z_{S_i}\|_2 \le \sqrt{2}/N = c_i$. One also has $\mathbb{E}(\|z - z_S\|_2^2) \le 1/N$ because of the variance of a multinomial, and thus $\mathbb{E}(\|z - z_S\|_2) \le \sqrt{\mathbb{E}(\|z - z_S\|_2^2)} \le 1/\sqrt{n}$.

Applying the McDiarmid's inequality gives $\mathbb{P}(\|z_s - z\|_2 \ge \mathbb{E}(\|z - z_S\|_2) + \epsilon) \le e^{-\frac{\epsilon^2}{2 \sum c_i^2}}$. With $\delta = e^{-\frac{\epsilon^2}{2 \sum c_i^2}} = e^{-\frac{N\epsilon^2}{4}}$, thus $\epsilon = \sqrt{\frac{2 \log(1/\delta)}{N}}$, one has the result. $\qquad\square$

### 7.3.1 Proof of the Proposition 5

Let $\mu_2$ be as in Lemma 7. By the Lemma 9, with probability $1 - \delta$, one has $\mathcal{H}_0 \subset \mathcal{H}_\mu^S$, thus $rank(H) \le d$ for any $H \in \mathcal{H}_\mu^S$. Moreover, as $\mathcal{H}_\mu^S \subset \mathcal{H}_{2\mu}$, the condition $\mu < \mu_2$ implies that $rank(H_\varepsilon) \ge d$ for any $H \in \mathcal{H}_\mu^S$. $\qquad\square$

### 7.4 Proof of the Proposition 6

**Lemma 10.** *Let $p$ be a rank $d$ distribution computed by an FST. Let $S$ be an i.i.d. sample of size $N$ with respect to $p$. Let $\delta > 0$ be a confidence parameter. For any $\epsilon > 0$, let $\mu_\epsilon$ be as in Lemma 8. One supposes that*

$$N > \left(\frac{1 + \sqrt{2 \log(1/\delta)}}{\mu_\epsilon}\right)^2$$

*With probability $1 - \delta$, for any $H \in \mathcal{H}_{\mu_\epsilon}^S$, $min_{H_0 \in \mathcal{H}_0}(\|H - H_0\|_F) < \epsilon$.*

*Proof.* This is just Lemma 8 and Lemma 9 together. $\qquad\square$

Let us define the distance between two models with the same rank:

**Definition 16.** *Let $A = (\alpha_1, \alpha_\infty, M_y^x)$ and $A' = (\alpha_1', \alpha_\infty', M_y'^x)$ be two FSTs with $d$ states, on the same alphabet. On defines the distance*

$$|A, A'|_\infty = \max\left(\max_i(|(\alpha_1)_i - (\alpha_1')_i|), \max_i(|(\alpha_\infty)_i - (\alpha_\infty')_i|), \max_{i,j,x,y}(|(M_y^x)_{i,j} - (M_y'^x)_{i,j}|)\right)$$

Let us recall a result [12]:

**Lemma 11.** *Let $H$ and $H' = H + E$ be two $n \times m$ matrices. Let $\sigma_1 \ge \cdots \ge \sigma_n$ be the singular values of $H$, and let $\sigma_1' \ge \cdots \ge \sigma_n'$ be the singular values of $H'$. One then has*

$$|\sigma_i - \sigma_i'| \le \|E\|_2$$

Let $H = L^\top D R$ and $H' = L'^\top D' R'$ be the singular value decompositions of $H$ and $H'$. One has $H^+ = R^\top D^{-1} L$ and $H'^+ = R'^\top D'^{-1} L'$. One has:

**Lemma 12.** *Let $H$ and $H' = H + E$ be two $n \times m$ matrices. Let $H = L^\top D R$ and $H' = L'^\top D' R'$ be the singular value decompositions of $H$ and $H'$. Let $\sigma$ be such that $\forall i, \sigma_i \geq \sigma, \sigma'_i \geq \sigma$. One has*

$$\|D^{-1} - D'^{-1}\|_F \leq \|D^{-1} - D'^{-1}\|_* \leq \frac{d\|E\|_2}{\sigma^2}$$

*Proof.* On has $|\frac{1}{\sigma_i} - \frac{1}{\sigma'_i}| \leq |\frac{\sigma'_i - \sigma_i}{\sigma_i \sigma'_i}| \leq \frac{\|E\|_2}{\sigma^2}$, and one has the conclusion. $\qquad\square$

The following result is straightforward from [19]:

**Lemma 13.** *Let $H$ and $H' = H + E$ be two matrices. Let $\sigma_1 \geq \cdots \geq \sigma_n$ be the singular values of $H$, and let $\sigma'_1 \geq \cdots \geq \sigma'_n$ be the singular values of $H'$. Let $\sigma$ be such that $\forall i, \sigma_i \geq \sigma, \sigma'_i \geq \sigma$. Let $H = L^\top D R$ and $H' = L'^\top D' R'$ be the singular value decompositions of $H$ and $H'$. One supposes that $\|E\|_F \leq \sigma/2$. One then has*

$$\|L - L'\|_F \leq \frac{4(2\sqrt{d}\|H\|_F\|E\|_F + \|E\|_F^2)}{\sigma^2}, \|R - R'\|_F \leq \frac{4(2\sqrt{d}\|H\|_F\|E\|_F + \|E\|_F^2)}{\sigma^2}$$

### 7.4.1 Proof of Proposition 6

Let $\mu_\epsilon$ be as in Lemma 8. The condition on $N$ implies $\mu < \mu_\epsilon$. Let $H \in \mathcal{H}_\mu^S$, there exists $H' \in \mathcal{H}_0$ such that $\|H - H'\|_F < \epsilon$. One has $\|L\|_F = \|L'\|_F = \|R\|_F = \|R'\|_F = \sqrt{d}$, as the matrices are orthonormal. One has also $\|D^{-1}\|_F \leq \sqrt{d}/\sigma$. One uses the equality $AB - A'B' = (A - A')B - (A - A')(B - B') + A(B - B')$. One has

$$H^+ - H'^+ = L^\top D^{-1} R - L'^\top D'^{-1} R'$$

$$= L^\top[(D^{-1} - D'^{-1})R - (D^{-1} - D'^{-1})(R - R') + D^{-1}(R - R')]$$

$$-(L^\top - L'^\top)[(D^{-1} - D'^{-1})R - (D^{-1} - D'^{-1})(R - R') + D^{-1}(R - R')] + (L^\top - L'^\top)D^{-1}R$$

Using the previous inequalities, and keeping only the first order terms, leads to

$$\|H^+ - H'^+\|_F \leq O(\frac{d^2\epsilon}{\sigma^3})$$

One also has $\|H^+\|_F \leq \frac{d^2}{\sigma}$. Plugging all those inequalities in the formulas computing the FSTs parameters leads to the result. $\qquad\square$