[Reviews · NeurIPS 2013]

Submitted by Assigned_Reviewer_2

Overview: This paper looks at the difficult problem of learning FST models of unaligned input and output sequences. This is an interesting problem and the approach appears to have merit; the main drawback of the paper is that some sections are very difficult to understand.

Major Comments:
1. In the abstract and introduction: the authors mention that the setting where sequences are not aligned is more realistic. I believe the authors, but examples of problems where unaligned sequences are the norm would be welcome.
2. This was a very dense paper that took me quite a while to understand. In particular the critical Sections 2.3 and 3.1 are close to impenetrable. I think it would be helpful to expand these sections while maybe cutting Section 4.1 up to example 4 in order to make the necessary room.
3. The experimental results do seem to show that the proposed method does very well at recovering a FST model with the unsupervised approach while missing alignment information on a simple problem. However, given that in the introduction the authors claimed that the motivation for learning FST models of unaligned input and output sequences was biology and natural language processing (where unaligned input and output sequences are common), I was a bit disappointed that the experimental results were only conducted on a very simple toy problem.


Minor Comments:
1. The figures after the first paragraph (DNA sequences?) need some context or caption.
2. Line 67 “constraint” → “constrain”
3. It is not possible to tell apart the red and blue lines in a non-color printout in Example 1 .
4. The text in Figure 1 is way too small.
Summary: The paper contains an interesting algorithm to solve an interesting problem, but is also very dense and difficult to understand.

Submitted by Assigned_Reviewer_5

This paper presents a spectral algorithm for learning FSTs in an unsupervised
manner (i.e. without observing the input-output alignments). This is done by
introducing a generalized Hankel matrix and deriving a result (Prop. 4) that
extends a previous result for the case where the alignments were observed.
The Hankel matrix is estimated by formulating a constrained rank optimization
problem, which is then relaxed to minimizing the nuclear norm.

Overall, this is a nice paper that extends previous work on spectral learning
of FSTs, the novelty being in addressing the case of unobserved alignments,
which complicates the problem quite a bit. The paper is clearly written and
everything is defined and stated rigorously.

The theoretical properties in 4.1 are nice to have, but they all refer to the
solution of the NP-hard problem so are somewhat useless. Can we claim something
about the solution of the relaxed problem?

In the experiments, it's a little disappointing that the EM algorithm performed
better than the spectral learning methods (but the same has happened in prior
work on spectral learning).

Minor comments:
- In Def. 4, shouldn't the empty-empty alignment {(eps_+, eps_-)} be excluded from \Sigma?
- In Prop. 2, the sum \sum_{s \in \Sigma_+, t \in \Sigma_-} should be
\sum_{s \in \Sigma_+^*, t \in \Sigma_-^*} (i.e., the Kleene stars are missing)
- line 154: a punctuation sign is missing before "One"
- line 161: "On" -> "One"
- in def. 8: "\in V_0" -< "v \in V_0" (i.e., v is missing)
- in the first paragraph of 3, p(s,t) is used without being introduced. Are these
empirical probabilities?
Summary: This paper presents a spectral algorithm for learning FSTs in an unsupervised
manner (i.e. without observing the input-output alignments). The Hankel matrix is estimated by formulating a constrained rank optimization problem, which is then relaxed to minimizing the nuclear norm.

Pros:
- extends previous work on spectral learning of FSTs by addressing the case of unobserved alignments
- some theoretical results regarding the constrained rank problem

Cons:
- those theoretical properties are not applicable to the relaxed problem which is actually solved

Submitted by Assigned_Reviewer_6

This is a rather technical paper that introduces a convex optimization-based (spectral) formulation for unsupervised learning of finite state transducers.

Unlike previous spectral approaches, the proposed algorithm does not require explicit alignment information for the input-output pairs in the training data. This causes difficulties in that the Hankel matrix typically used in spectral learning is not fully observed.

The main idea of the paper (which was presented before in [3], but in the slightly different setup of missing matrix values due to distribution mismatch), is that instead of using the full matrix, observable statistics can be computed instead that constrain the coefficients of the Hankel matrix. For instance, it is known that the probability of observing a a pair of unaligned input-output sequences is computed by summing over all possible alignments.

The main technical contributions of the paper are sample complexity bounds as well as identifiability results for the proposed estimation problem.

The experiments are synthetic and do not reveal a lot about the practical viability of the proposed approach. An EM-based approach is shown to outperform the proposed approach in terms of L1 error, but claimed to be more expensive computationally. Actual numbers regarding the computational efficiency of the considered algorithms are not given in the paper.

Quality:
The proposed approach is backed up by extensive theory. Whether or not the approach is useful in practice is hard to tell from the paper. In this reviewer's opinion, it would have substantially strengthened the submission if more practical experiments had been included in the manuscript. Clearly, FSTs are a very practical construct with numerous applications in NLP and other disciplines, so there is no good excuse for not including such experiments. As it stands, I have to assume that the approach either does not scale to practical amounts of data, or has some other limitations that keep it from being applied to real data.

Clarity:
Owing to the technical nature of the contents, the manuscript is not an easy read. However, the paper is well-written in general and I recognize that there is no magical way of presenting formal results in an easily comprehensible manner.
In any case, the presentation is clear enough to allow for implementation of the learning algorithm, although the experiments section probably lacks sufficient detail in order to allow for reproduction of the results.

Originality:
The proposed approach follows the same main idea that was already presented in [3]; as such, I consider the present work to be mostly incremental in nature.

Significance:
Unsupervised training of finite state transducers is an important problem that attracts a lot of attention in the natural language processing community, and deservedly so. Significant advances in this area are bound to have a major impact. However, from the present manuscript, it is not entirely clear whether the proposed approach represent such an advance.
Summary: This is a highly technical but rather well-written paper that introduces a spectral approach to unsupervised learning of finite state transducers. The authors derive several interesting theoretical properties of their approach (such as sample complexity bounds and identifiability results), but the practical relevance is not demonstrated convincingly.
Author Feedback

Author rebuttal: We thank the reviewers for their valuable feedback.

All reviewers have pointed out that our experiments are somewhat limited. We agree that empirical validation of the method in real transduction tasks is an important research goal. To achieve that goal we would first need to scale the optimization so that it can handle larger problems. This requires an additional engineering effort that we leave outside of the scope of this contribution. In any case, we believe that the engineering challenge can be solved effectively, since recently there have been large-scale applications of matrix completion techniques (e.g. netflix challenge) where the underlying optimization problem is similar in nature to ours. That been said, doing a good empirical study on a real task is probably a full paper by itself.

Based on the scores, all reviewers have judged our work as "Incremental and unlikely to have much impact". We respectfully disagree with the "incremental" assessment, which typically has a negative implication in the ML community. It is true that we study the same family of FST models than Balle et al ECML-2011, who proposed a spectral method. However, the differnces in the assumptions made by the learning algorithm are far from superficial (as some reviewers have actually pointed out). Spectral methods until now have been mainly applied to problems in which the unobserved variables are restricted to be the
hidden states. In this paper we study the more fundamental issue of missing state and structure, which leads to problems where the Hankel matrix is not directly observable. The main contribution of the paper is a new spectral approach that allows the recovery of structure and missing state from observable statistics. Although superficially this
might seem a small difference, in reality it is a fundamental one. This is because our approach opens the door for the application of spectral methods to a wide range of problems in which both state and structure are unobservable. For example, we are currently working on an application of this method to learn Weighted Context-free Grammars from string statistics only, and the main ingredients of the method remain the same. We are framing FST and grammar induction as a low-rank matrix optimization, and as far as we can see this is radically different from any existing method.


Reviewer 1

With regards to clarity, (as the other two reviewers acknowledged) it is hard to present the main results of the paper in a way that it will make it easily accessible to a wide audience. We will try to include more intuitions about the method wherever is possible, and perhaps follow the suggestion of dropping one example. For examples of problems where alignments are not available consider: speech recognition, phonetic-transliteration, machine translation, video event
recognition, etc.

Reviewer 2

Reviewer 2 questions the value of the bounds given for the rank minimization problem (since the problem itself is NP-hard). The theoretical results on the rank minimization problem prove that the target distributions are identifiable and that the problem is statistically consistent. Thus it establishes that the difficulty of learning FSTs is in essence computational. With regards to the applicability of the bounds to the relaxation: It is known that for
problems satisfying certain properties (RIP or NSP, cf http://arxiv.org/abs/1205.2081, and http://arxiv.org/abs/1204.1580)) the solution of the relaxed problem is a solution to the rank-minimization problem. Unfortunately, it is also known that checking for these properties is NP-hard. In practice the nuclear norm relaxation seems to give very good solutions. Clearly, when the solution of the relaxed problem happens to be a solution of the rank minimization problem the bounds apply.


Reviewer 3

With respect to actual costs of the experiments. We used Matlab, and we used the CVX optimization package in order to implement the convex optimization. For the comparison with EM reported in Table (c), the convex optimization took roughly 1400 seconds, while EM took roughly 150 seconds per iteration and required at least 100 iterations to get to the reported performance. Hence, under our current implementation,
the convex optimization is about 10 times faster. We can report these figures in the next version. However, we are reluctant to take any conclusion from these runs, as we have not identified the most efficient method to solve the convex optimization, and our implementations have not been optimized either. This engineering effort would allow us to have a proper empirical study on real datasets.